# Applicability of sentinel lymph node oriented treatment strategy for gallbladder cancer

**Koya Yasukawa**[ID]°*, **Akira Shimizu**°, **Hiroaki Motoyama**‡, **Koji Kubota**‡, **Tsuyoshi Notake**‡, **Shinsuke Sugenoya**‡, **Kiyotaka Hosoda**‡, **Hikaru Hayashi**‡, **Ryoichiro Kobayashi**‡, **Yuji Soejima**

Division of Gastroenterological, Hepato-Biliary-Pancreatic, Transplantation and Pediatric Surgery, Department of Surgery, Shinshu University School of Medicine, Matsumoto, Japan

° These authors contributed equally to this work.
‡ These authors also contributed equally to this work.
* kouyayasu@shinshu-u.ac.jp

## Abstract

### Background

Utility of the sentinel lymph node (SLN) biopsy in some malignancies has been reported, however, research on that of gallbladder cancer (GBC) is rare. The aim of this study is to investigate whether the concept of SLN is applicable to T2/3 GBC.

### Methods

A total of 80 patients who underwent resection for gallbladder cancer were enrolled in this study. Patients with GBC were stratified into two groups based on the location of tumor, peritoneal-side (T2p or 3p) and hepatic-side (T2h or 3h) groups. We evaluated the relationship between cystic duct node (CDN) and downstream lymph node (LN) status. CDN was defined as a SLN in this study.

### Results

Thirty-eight patients were classified into T2, including T2p (n = 18) and T2h (n = 20), and 42 patients into T3, including T3p (n = 22) and T3h (n = 20). The incidence of LN metastasis was significantly higher in hepatic-side than peritoneal-side in both T2 and T3 (P = 0.036 and 0.009, respectively). In T2, 14 T2p had negative CDN and downstream LN, however, three T2h had negative CDN and positive downstream LNs (defined as a skipped LN metastasis) (P = 0.043). In T3, patients with skipped LN metastasis were significantly higher in T3h (n = 11) than those in T3p (n = 2) (P<0.001). There was no recurrence of the local lymph node. Disease-free survival in the T2p and T3p were significantly better than those in the T2h and T3h (P = 0.005 and 0.025, respectively).

### Conclusion

The concept of SLN can be applicable to T2p GBC, where the downstream LNs dissection can be omitted.

**Data Availability Statement:** Data cannot be shared publicly because of the point of view of personal information protection. Data are available from the Shinshu Institutional Data Access / Ethics

Committee (contact via shinhp@shinshu-u.ac.jp) for researchers who meet the criteria for access to confidential data. The data underlying the results presented in the study are available from (Shinshu University: shinhp@shinshu-u.ac.jp).

**Funding:** This research did not receive any specific grant from funding agencies in the public, commercial, or not-for-profit sectors.

**Competing interests:** The authors have declared that no competing interests exist.

## Introduction

Gallbladder cancer (GBC) has a very poor prognosis, except for early stage cancer. Complete surgical resection is the only potential way to achieve long-term survival [1, 2]. One of the most important factors for improving the prognosis of GBC is lymph node dissection (LND), and LND plays a crucial role in terms of reducing the risk of recurrence from remnant lymph node (LN) metastasis [3]. However, indication and extent of LND is still controversial [2]. Furthermore, extensive LND may be associated with intra- and postoperative complications.

A sentinel lymph node (SLN) biopsy is a removal of draining LNs that are deemed likely to first receive lymph flow from the area of the resected organ and is examined by a pathologist to determine the presence of metastasis. This has been proposed as a technique to identify LN metastases while reducing operative complications associated with aggressive LND [4]. According to the recent reports, intraoperative pathological examination of the SLN has been performed some malignancies [5–8]. In fact, SLN biopsy is clinically used to determine the extent of LND in breast cancer and malignant melanoma, however, there are no reports regarding GBC.

Some researchers [9–11] reported that there were two major pathways for lymph drainage (the left oblique pathway to the celiac nodes and the right descending pathway to the superior retropancreaticoduodenal node) and one minor pathway for lymph drainage (the superior mesenteric nodes). Furthermore, Uesaka et al. reported [12] that these pathways passed through the gallbladder neck LN as Calot's node without exception. If SLNs are present in GBC, they are likely to be this Calot's node.

Accordingly, we hypothesized that the concept of SLN is also applicable to GBC. To testify this hypothesis, we analyzed pathological LN status of GBC and whether the determination of the extent of LND is possible by SLNs biopsy in T2 and T3 GBC.

## Materials and methods

### Patients

In this study, 106 consecutive patients who underwent surgical resection for T2 or T3 GBC at Shinshu University Hospital between March 1990 and September 2018 were included. Their medical records were reviewed retrospectively. We excluded patients who did not receive LND (n = 4), patients who did not have data on location or had missing data on location (n = 6), patients who underwent incomplete resection (R1 and R2 resection) (n = 15) and patients with a pathological diagnosis of squamous cell carcinoma (n = 1). All patients underwent radiological examinations including computed tomography, magnetic resonance imaging, ultrasonography, and endoscopic ultrasonography to determine preoperative T stage. The pathologic tumor-node-metastasis (TNM) stage was defined according to the AJCC guidelines (8th edition) [13]; where T2p and T3p were defined as peritoneal-side and T2h and T3h as hepatic-side. If the tumor was located at the transition between the peritoneal-side and hepatic-side, T stage was classified as T2h or T3h (hepatic-side) (Fig 1A). Finally, a total of 80 patients were included in this study. 38 patients were classified into T2 GBC, including T2p (n = 18) and T2h (n = 20), and 42 patients into T3 GBC, including T3p (n = 22) andT3h (n = 20).

### Lymph node dissection

The range of LN dissection is determined by reference to the AJCC guidelines (8th edition) [13]. The cystic duct, porta hepatis, hepatoduodenal ligament, superior pancreaticoduodenal, and common hepatic artery LNs were routinely harvested for T2 and T3 GBC (Fig 1B), except for the paraaortic LN.

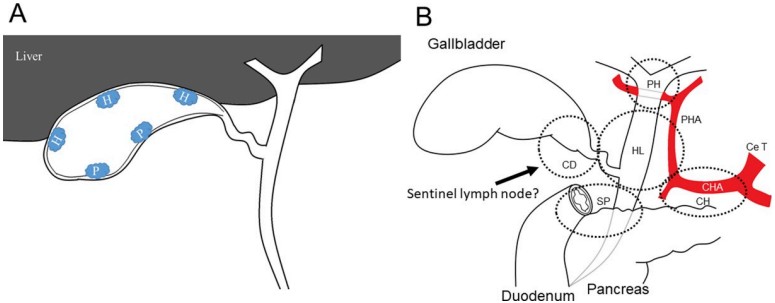

**Fig 1. A.** Scheme of tumor location. When tumor was on the peritoneal-side, it was classified as T2p, and when tumor was on the hepatic-side, it was classified as T2h. In addition, it was classified as T2hwhen tumor was located in the transition between the peritoneal-side and hepatic-side. H, hepatic-side; P, peritoneal side. **B.** Range for lymph node dissection. Ce T, celiac trunk; CHA, common hepatic artery; PHA, proper hepatic artery; CD, CDN; PH, porta hepatis; HL, hepatoduodenal ligament; SP, superior pancreaticoduodenal; CH, common hepatic artery lymph nodes.

## Surgical procedure

All patients with more than T2 GBC underwent laparotomy. In T2p, simple cholecystectomy with LND, bile duct resection (BDR), and gallbladder bed resection (GBR) were performed, while LND, BDR, GBR, and/or extended right lobectomy (Hx) were done in T2h if right hepatic artery was infiltrated. In T3, BDR, GBR, Hx, pancreaticoduodenectomy (PD) and/or hepatectomy with concomitant PD (HPD) were determined based on the preoperative radiological examinations and the intraoperative findings. All patients who underwent Hx or HPD received portal vein embolization (PVE) before surgery (planned HPD).

## Postoperative follow-up

After discharge, the patients were followed up every 2–3 months with ultrasonographic examination in our outpatient clinic. Computed tomography or magnetic resonance imaging was performed every 6 months, or as necessary. Recurrence was detected by imaging findings.

After April 2010, patients received adjuvant chemotherapy except those with poor performance status or who refused chemotherapy. Gemcitabine hydrochloride alone (GEM; 2010–2015), tegafur/gimeracil/oteracil (S-1; after 2015) or GEM plus S-1 or GEM plus cisplatin (CDDP) combination therapies (after 2015) were administered as per each patient's condition. Patients with recurrence after surgical resection received GEM alone, GEM plus S-1 combination, or GEM plus CDDP combination chemotherapy according to each condition.

## Definition

Positivity for downstream LNs, despite being negative for cystic duct node (CDN), was defined as 'skipped LN metastasis'. Surgical mortality was defined as intraoperative death, death within 30 days after surgery, and in-hospital death. Major complications were defined as grade III–IV events according to the Clavien–Dindo classification [14].

## Ethics statements

This study was approved by the ethics committee of Shinshu University School of Medicine (approval no. 2020–4558), and the investigation was conducted according to the principles expressed in the Declaration of Helsinki. All patients were provided with complete information about the study and provided their consent for participation, and written informed consent was obtained from all patients before enrolment. The data were analyzed retrospectively

and anonymously on the basis of medical records, and the authors did not have access to identifying patient information or direct access to the study participants.

## Statistical analysis

Data were collected retrospectively for all participating patients; these data included patient preoperative blood exam data, surgical outcomes, postoperative complications, and radiological examinations. Continuous data are expressed as median values (range), unless stated otherwise. We compared continuous variables using the Mann–Whitney U test; categorical variables were compared using the $\chi^2$ test or Fisher's exact test. Disease-free survival (DFS) was analyzed by the log-rank test and plotted by the Kaplan–Meier method. DFS was defined as the time from surgery to recurrence, death from any cause, or the final follow-up. P values of 0.050 were considered statistically significant. Statistical analyses were performed using JMP® 14 (SAS Institute Inc., Cary, NC, USA).

## Results

### Characteristics and outcomes according to the tumor location

The clinicopathological characteristics and surgical outcomes of patients with T2 or T3 GBC are shown in Table 1. The factors of age, sex, preoperative carcinoembryonic antigen and carbohydrate antigen 19–9 were comparable. In T2 and T3 patients, GBR were higher in T2/3h tumor (n = 17 and 16) than that in T2/3p (n = 7 and 7) (P = 0.006 and 0.029, respectively), while the other surgical procedures were not significantly different including HPD. Four of the 18 T2p patients and 11 of the 20 T2h patients had LN metastasis. The incidence of LN metastasis was significantly higher in T2h compared with T2p patients (P<0.001). In T3 patients, nine of the 22 T3p patients had LN metastasis, compared with 16 of 20 T3h patients. Furthermore, the incidence of postoperative complications differed significantly between T3p and T3h (0.011). Occurrence of posthepatectomy biliary leakage (T3p: n = 2, T3h: n = 7, P = 0.021) and cholangitis (T3p: n = 0, T3h: n = 4, P = 0.038) was significantly higher in T3h than that in T3p, however, no significant difference was observed in the Clavien–Dindo classification[19] grade ≥ III complication between the groups. Throughout this study, the operative mortality was 0%, and posthepatectomy biliary leakage occurred in three T2 (7.8%) and eight T3 (20%) patients. Other complications consisted of surgical site infection (n = 5), pleural effusion requiring puncture (n = 8), cholangitis (n = 4), intraabdominal abscess (n = 4), and postoperative pancreatic fistula (n = 6). Postoperative bleeding, as a Clavien–Dindo classification[19] grade IIIb complication, was observed in one patient, who underwent reoperation. No significant difference was observed in the adjuvant chemotherapy between the groups.

When DFS was compared between peritoneal-side and hepatic-side tumor location in T2, DFS in the T2p was significantly better than that in the T2h (5-year DFS: 77.8% and 50.0%, respectively; P = 0.005) (Fig 2A). Likewise, the difference for DFS in T3p was significantly better than that in the T3h (5-year DFS: 40.9% and 15.0%, respectively; P = 0.025) (Fig 2B).

### Assessment for lymph node status

In this study, all patients received full LND including the CDN, hepatoduodenal ligament, porta hepatis, superior pancreaticoduodenal, and common hepatic artery LNs. The relationships between CDN and downstream LNs classified by tumor location are shown in Table 2, and the summary of location of LNs metastases are shown in Fig 3.

In T2 GBC, 14 T2p patients had negative CDN and downstream LNs, however, three T2h patients had skipped LN metastasis (P = 0.043). This skipped LN metastasis was more evident

**Table 1. The clinicopathological characteristics and surgical outcomes according to tumor location in patients with T2/3 gallbladder cancer.**

| | T2 (n = 38) | | | T3 (n = 42) | | |
|---|---|---|---|---|---|---|
| | T2p* | T2h* | P† | T3p | T3h | P† |
| Variable | (n = 18) | (n = 20) | value | (n = 22) | (n = 20) | value |
| Age | 68.0 ± 10.2 | 69.0 ± 9.8 | 0.671‡ | 73.5 ± 9.1 | 69.5 ± 9.2 | 0.384‡ |
| Sex (Male/Female) | 6/12 | 10/10 | 0.176 | 11/11 | 11/9 | 0.746 |
| Preoperative CEA (ng/mL) | 1.9 ± 5.4 | 2.7 ± 8.4 | 0.259‡ | 2.5 ± 21.0 | 2.25 ± 49.3 | 0.730‡ |
| Preoperative CA 19–9 (U/mL) | 14.2 ± 30.1 | 19.75 ± 38.9 | 0.275‡ | 2.5 ± 21.0 | 67.7 ± 975.4 | 0.701‡ |
| Surgical procedures | | | | | | |
| BDR | 14 (77.8) | 10 (50.0) | 0.101 | 21 (99.5) | 19 (95.0) | 0.989 |
| GBR | 7 (38.9) | 17 (85.0) | 0.006 | 7 (31.8) | 16 (80.0) | 0.029 |
| Hx | 0 (0.0) | 7 (29.2) | 0.087 | 9 (40.9) | 4 (20.0) | 0.190 |
| PD | 0 (0.0) | 0 (0.0) | NS | 8 (36.4) | 11 (55.0) | 0.352 |
| HPD | 0 (0.0) | 0 (0.0) | NS | 1 (4.5) | 4 (20.0) | 0.174 |
| Operation time** | 364.5 | 587.5 | 0.066‡ | 605.0 | 641.5 | 0.782‡ |
| (min) | (91–823) | (138–990) | | (120–1045) | (495–1125) | |
| Intraoperative bleeding** | 225 | 375 | 0.062‡ | 765 | 825 | 0.629‡ |
| (ml) | (0–800) | (20–2220) | | (90–3520) | (200–2470) | |
| Blood transfusion | 0 (0.0) | 5 (20.8) | 0.053 | 3 (13.6) | 5 (25.0) | 0.348 |
| Complications rate | 7 (38.9) | 10 (50.0) | 0.760 | 8(36.4) | 16 (80.0) | 0.011 |
| Tumor size (mm) | 31.5 ± 12.4 | 37.0 ± 11.9 | 0.294‡ | 41.9 ± 15.4 | 44.1 ± 14.1 | 0.645‡ |
| Lymph node metastasis* | 4 (22.2) | 11 (55.0) | 0.036 | 8 (36.4) | 16 (80.0) | 0.009 |
| Venous invasion* | 9 (50.0) | 11 (55.0) | 0.757 | 19 (86.4) | 18 (90.0) | 0.715 |
| Lymphatic vessel invasion* | 10 (55.5) | 15 (75.0) | 0.206 | 19 (86.4) | 18 (90.0) | 0.715 |
| Perineural invasion* | 5 (27.8) | 10 (50.0) | 0.159 | 16 (72.7) | 18 (90.0) | 0.146 |
| Histological grade* | | | 0.152 | | | 0.319 |
| G1/2 | 13 (72.2) | 17 (85.0) | | 15 (68.2) | 12 (60.0) | |
| G3 | 5 (27.8) | 3 (15.0) | | 7 (31.8) | 8 (40.0) | |
| Adjuvant chemotherapy | 0 (0.0) | 2 (10.0) | 0.842 | 5 (22.7) | 3 (15.0) | 0.348 |

Abbreviations: CEA = carcinoembryonic antigen; CA19–9 = carbohydrate antigen 19–9; BDR = bile duct resection; GBR = gallbladder bed resection; Hx = hepatectomy (extended right lobectomy); PD = pancreaticoduodenectomy; HPD = hepatectomy with concomitant PD.

*According to the definition of the American Joint Committee on Cancer Staging Manual, 8th edition.

Values in parentheses are percentages unless indicated otherwise;

** median (range).

†$\chi^2$ or Fisher's exact test, except

‡Mann–Whitney U test.

in both T3p and T3h, in which two of the 22 T3p patients showed skipped LN metastasis, as compared to 11 of the 20 patients with T3h ($P<0.001$).

Incidence of the common hepatic artery LN metastasis, which was furthest from the tumor, was significantly higher in T2h compared with T2p patients ($P = 0.019$). Likewise, porta hepatis and common hepatic artery LN metastases were significantly more common in T3h compared with T3p ($P = 0.005$ and 0.016, respectively).

## Recurrence site

Tumor recurrence was observed in 47/80 (58.8%) patients in entire population analysis (Table 3). In T2, tumor recurrence was observed in 4/18 (22.2%) patients with peritoneal-side tumor and 12/20 (60.0%) patients with hepatic-side tumor, while 13/22 (59.1%) and

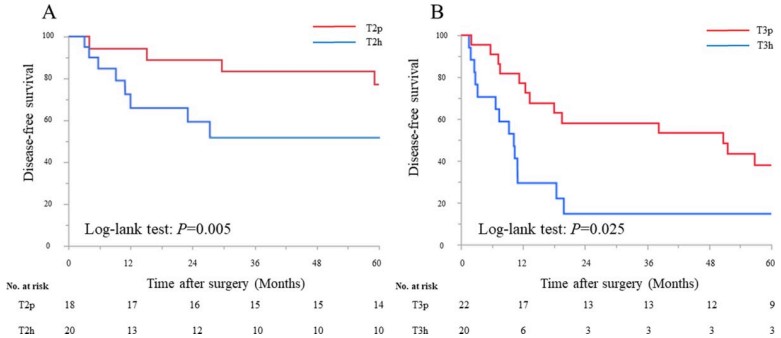

**Fig 2. Kaplan–Meier survival analysis according to the tumor location in T2 and T3. A.** Disease-free survival between tumors located on the peritoneal-side or hepatic-side in T2. **B.** Disease-free survival between tumors located on the peritoneal-side or hepatic-side in T3.

18/20 (90.0%) were observed in T3. The recurrence rate in patients with the tumors located on the hepatic-side was significantly higher than that in patients with tumor located on the peritoneal-side in both T2 and T3 ($P = 0.025$ and $0.035$, respectively). Specifically, the recurrence rate of liver metastasis in patients with the hepatic-side tumor was significantly higher the than that of patients with peritoneal-side tumor in both T2 and T3 ($P = 0.019$ and $0.023$, respectively). Furthermore, the recurrence in para-aortic lymph node metastasis was only observed in patients with hepatic-side tumor in T2, and there was significant difference ($P = 0.043$). However, no difference was observed in T3 ($P = 0.899$); peritoneal-side were three patients, hepatic-side were also three. In both groups, there was no recurrence of the local lymph node.

**Table 2. Relationship of lymph node metastasis, recurrence site and tumor location in patients with T2/3 gallbladder cancer.**

| | T2 (n = 38) | | | T3 (n = 42) | | |
|---|---|---|---|---|---|---|
| | **T2p** | **T2h** | **P†** | **T3p** | **T3h** | **P†** |
| | **(n = 18)** | **(n = 20)** | **value** | **(n = 22)** | **(n = 20)** | **value** |
| Lymph node metastasis | | | 0.036 | | | 0.009 |
| Positive | 4 (22.2) | 11 (55.0) | | 8 (36.4) | 16 (80.0) | |
| Negative | 14 (77.8) | 9 (45.0) | | 14 (63.6) | 4 (20.0) | |
| Location of lymph node metastasis | | | | | | |
| Cystic duct | 4 (22.2) | 8 (40.0) | 0.235 | 6 (27.3) | 5 (25.0) | 0.867 |
| Hepatoduodenal ligament | 4 (22.2) | 10 (50.0) | 0.072 | 8 (36.4) | 12 (60.0) | 0.124 |
| Porta hepatis | 4 (22.2) | 9 (45.0) | 0.135 | 6 (27.3) | 14 (70.0) | 0.005 |
| Superior pancreaticoduodenal | 3 (16.7) | 8 (40.0) | 0.108 | 5 (22.7) | 7 (35.0) | 0.379 |
| Common hepatic artery | 1 (5.6) | 7 (35.0) | 0.019 | 2 (9.1) | 8 (40.0) | 0.016 |
| Skipped lymph node metastasis* | 0 (0) | 3 (15.0) | 0.043 | 2 (9.1) | 11 (55.0) | <0.001 |
| Hepatoduodenal ligament | 0 (0) | 2 (10.0) | | 2 (9.1) | 7 (35.0) | |
| Porta hepatis | 0 (0) | 2 (10.0) | | 0 (0) | 10 (50.0) | |
| Superior pancreaticoduodenal | 0 (0) | 2 (10.0) | | 2 (9.1) | 5 (25.0) | |
| Common hepatic artery | 0 (0) | 1 (5.0) | | 1 (4.5) | 5 (25.0) | |

Values in parentheses are percentages.

*Skipped lymph node metastasis was defined as positivity for the downstream lymph nodes despite negative cystic duct lymph node.

†χ2 or Fisher's exact test.

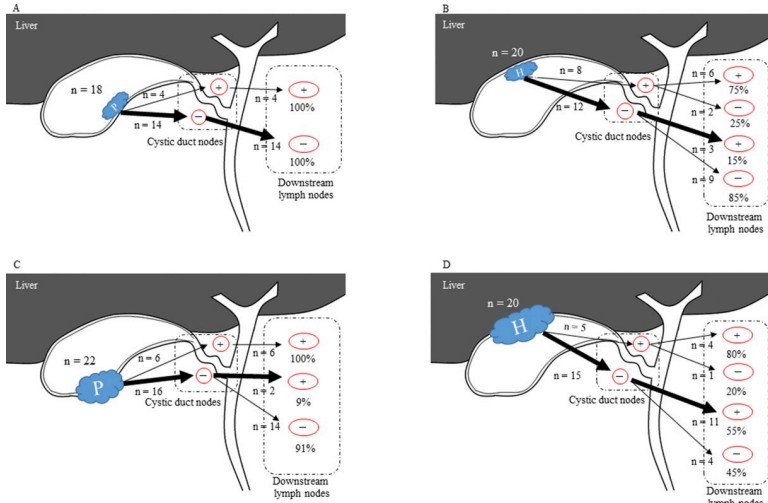

**Fig 3. Features of lymph node metastasis in T2 and T3 gallbladder cancer. A.** In T2p, if cystic duct lymph node was negative, downstream lymph nodes were also negative status. **B.** In T2h, even if CDN was negative, downstream lymph nodes were not negative. **C, D.** Even if CDN was negative, downstream lymph nodes were not negative. P, peritoneal-side; H, hepatic-side; +, lymph node positive; -, lymph node negative.

## Discussion

In this study, we demonstrated that if CDNs were negative, all downstream LNs were also negative in T2p patients. In contrast, if CDNs were positive, the downstream LNs were 100% positive. The sensitivity and specificity of the SLN as a test to determine whether the downstream LNs were positive or not were calculated as both 100% in T2p patients. On the other hand, sensitivity and specificity were 75.0% and 75.0% in T2h, 100% and 87.5% in T3p, and 80.0% and 26.7% in T3h patients, respectively. These results revealed that CDNs may be defined as a SLN in only T2p GBC. Furthermore, relatively high sensitivity and specificity were also observed for T2h and T3p.

Several reports [15–18] have demonstrated that the number of positive LNDs has been associated with prognosis, and proposed that at least 6 or more LN should be dissected. And recent reports have demonstrated that the extensive LND including posterosuperior pancreatic head LNs should be performed [19, 20]. Especially among them, Only Vega et al. [21] reported that

**Table 3. Relationships between the recurrence site and tumor location in T2/3 patients.**

| | T2 (n = 38) | | | T3 (n = 42) | | |
|---|---|---|---|---|---|---|
| | **T2p** | **T2h** | **P† value** | **T3p** | **T3h** | **P† value** |
| | **(n = 18)** | **(n = 20)** | | **(n = 22)** | **(n = 20)** | |
| Recurrence | 4 (22.2) | 12 (60.0) | 0.025 | 13 (59.1) | 18 (90.0) | 0.035 |
| Site of recurrence (duplicated) | | | | | | |
| Para-aortic lymph node | 0 (0.0) | 3 (15.0) | 0.043 | 3 (13.6) | 3 (15.0) | 0.899 |
| Liver metastasis | 1 (5.6) | 7 (35.0) | 0.019 | 3 (13.6) | 9 (45.0) | 0.023 |
| Locoregional | 1 (5.6) | 3 (15.0) | 0.332 | 2 (9.1) | 3 (15.0) | 0.554 |
| Dissemination | 1 (5.6) | 1 (5.0) | NA | 4 (18.2) | 6 (30.0) | 0.369 |
| Lung | 1 (5.6) | 0 (0.0) | 0.487 | 1 (0.0) | 2 (10.0) | 0.598 |

Values in parentheses are percentages.

†χ2 or Fisher's exact test.

the status of the CDN can predict the status of the hepatic pedicle nodes but not the presence or absence of more advanced LN metastasis. However, this report did not mention about relationship between LN metastasis and tumor location.

The important point requires clarification is why CDNs may be defined as a SLN in only T2p GBC. Some researchers [11–13] and Uesaka et al. [12] reported that there were three lymphatic pathways passed through the CDN without exception for GBC. However, since there was skipped LN metastasis in this study, there was another route for gallbladder lymphatics or a route though the hepatic-side. Some studies have reported that there are three types of spread of GBC [22]; LN metastasis through lymphatic invasion, tumor spread via lymphatic flow along the Glissonian pedicles, and hematogenous liver metastasis through the cystic veins. Therefore, when considering LN metastasis for GBC, in addition to the three pathways, the route through the liver must take into account. However, this study revealed that the route may not need to be considered only in T2p patients.

Furthermore, this study showed the differences in the form of recurrence and prognosis according to the tumor locations. Previous publications that describe the surgical outcomes difference of tumor location are summarized in Table 4 [23–30]. Except reported by Jung et al.

**Table 4. Previous reports on surgical outcomes according to the tumor location in gallbladder cancer.**

| No. | Author | Year | Location (side) | 5-year Survival rate (%) | Comparison*† | *P* value |
|---|---|---|---|---|---|---|
| 1 | Shindo et al. [23] | 2015 | Peritoneal, n = 153 | 64.7 (OS) | T2p vs **T2h** | <0.001 |
| | | | Hepatic, n = 99 | 42.6 (OS) | | |
| | | | Peritoneal, n = 136 (N0) | 66.0 (OS) | T2p vs **T2h** | |
| | | | Hepatic, n = 66 (N0) | 52.4 (OS) | | 0.040 |
| | | | Peritoneal, n = 88 | 25.0 (OS) | T3p vs T3h | |
| | | | Hepatic, n = 43 | 29.0 (OS) | | 0.610 |
| 2 | Lee et al. [24] | 2015 | Peritoneal, n = 33 | 96.0 (OS) | T2p vs **T2h**† | 0.007 |
| | | | Hepatic, n = 124 | 62.7 (OS) | | |
| 3 | Jung et al. [25] | 2016 | Peritoneal, n = 26 | 64.5 (DFS) | T2p vs T2h | 0.983 |
| | | | Hepatic, n = 62 | 65.2 (DFS) | | |
| 4 | Lee et al. [26] | 2017 | Peritoneal, n = 99 | 84.9 (OS) | T2p vs **T2h** | 0.048 |
| | | | Hepatic, n = 93 | 71.8 (OS) | | |
| 5 | Wang et al. [27] | 2018 | Peritoneal, n = 46 | N/A (OS) | T2p vs **T2h** | 0.041 |
| | | | Hepatic, n = 36 | N/A (OS) | | |
| 6 | Park et al. [28] | 2018 | Peritoneal only, n = 35 | N/A (DFS) | T2p vs **T2h** | 0.043 |
| | | | Hepatic and Peritneal, n = 36 | N/A (DFS) | | |
| 7 | Cho et al. [29] | 2019 | Peritoneal, n = 37 | N/A (OS) | T2p vs **T2h** | 0.041 |
| | | | Hepatic, n = 44 | N/A (OS) | | |
| 8 | Kim et al. [30] | 2020 | Peritoneal, n = 82 | 96.8 (OS) | T2p vs **T2h** | 0.007 |
| | | | Hepatic, n = 50 | 80.7 (OS) | | |
| 9 | Present | | Peritoneal, n = 18 | 77.8 (DFS) | T2p vs **T2h** | 0.005 |
| | | | Hepatic, n = 20 | 50.0 (DFS) | | |
| | | | Peritoneal, n = 22 | 40.9 (DFS) | T3p vs **T3h** | 0.025 |
| | | | Hepatic, n = 20 | 15.0 (DFS) | | |

Abbreviations: OS = overall survival; DFS = disease-free survival; p = peritoneal-side tumor; h = hepatic-side tumor; N/A = not applicable.

*Significantly worse groups are shown in bold underlined text.

†According to the definition of the American Joint Committee on Cancer Staging Manual, 8th edition (T2).

[25], other reports demonstrated that OS or DFS in the patients with tumors located on the hepatic-side were significantly worse than that in the tumors located on the peritoneal-side in T2. Although there were few reports of a difference in OS or DFS according to tumor location in T3, Shindoh et al. [23] had demonstrated that no difference was observed in OS between tumor location in T3. However, our study demonstrated that DFS in the tumor located on the peritoneal-side was significantly better than that in the hepatic-side in both T2 and T3. Furthermore, since all reports had revealed that lymph node metastasis was an independent poor prognostic factor in T2 GBC patients by multivariable analysis, research into how GBC metastases through lymph nodes and the extent of LND are important for improvement of surgical and oncological outcomes for GBC.

The present study had several limitations. First, this was a single-center retrospective study and may include a selection bias. Second, this study was conducted in a relatively small number of cases. Thus further study incorporating a large number of patients should be warranted to confirm our conclusions. Despite these drawbacks, we believe that our findings are of interest because to the best of our knowledge, no reports have demonstrated the utility of the SLN in GBC.

In summary, this study provides that if CDN, or defined as a SLN, had negative status, the downstream LNDs can be omitted in T2p patients with GBC. Furthermore, understanding the detailed mechanisms of how skip LN metastasis occurred could enable the basic research that leads to find accurate lymphatics pathway for GBC and new treatment options.

## Acknowledgments

We thank Renee Mosi, PhD, from Edanz Group (https://en-author-services.edanzgroup.com/) for editing a draft of this manuscript. This study was not preregistered in an independent institutional registry.

## Author Contributions

**Conceptualization:** Koya Yasukawa.

**Data curation:** Koya Yasukawa, Akira Shimizu, Hiroaki Motoyama, Koji Kubota, Tsuyoshi Notake, Shinsuke Sugenoya, Kiyotaka Hosoda, Hikaru Hayashi, Ryoichiro Kobayashi, Yuji Soejima.

**Formal analysis:** Koya Yasukawa, Akira Shimizu, Hiroaki Motoyama, Koji Kubota, Tsuyoshi Notake, Shinsuke Sugenoya, Kiyotaka Hosoda, Hikaru Hayashi, Ryoichiro Kobayashi, Yuji Soejima.

**Funding acquisition:** Koya Yasukawa.

**Investigation:** Koya Yasukawa.

**Methodology:** Koya Yasukawa.

**Project administration:** Koya Yasukawa.

**Resources:** Koya Yasukawa.

**Software:** Koya Yasukawa.

**Supervision:** Koya Yasukawa.

**Validation:** Koya Yasukawa.

**Visualization:** Koya Yasukawa.

**Writing – original draft:** Koya Yasukawa.

**Writing – review & editing:** Koya Yasukawa.

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
