## [Decision Letter · Decision Letter 0]

31 Dec 2020

PONE-D-20-32884

Applicability of Sentinel Lymph Node Oriented Treatment Strategy for Gallbladder Cancer

PLOS ONE

Dear Dr. Yasukawa,

Thank you for submitting your manuscript to PLOS ONE. After careful consideration, we feel that it has merit but does not fully meet PLOS ONE’s publication criteria as it currently stands. Therefore, we invite you to submit a revised version of the manuscript that addresses the points raised during the review process.

A few minor revisions are noted, mostly in terms of clarifying language. These are noted clearly in the reviewer's comments provided to you. Please submit your revised manuscript by January 31, 2021. If you will need more time than this to complete your revisions, please reply to this message or contact the journal office at plosone@plos.org. Please include the following items when submitting your revised manuscript:

We look forward to receiving your revised manuscript.

Kind regards,

Mitesh J Borad

Academic Editor

PLOS ONE

Additional Editor Comments:

The manuscript - "Applicability of Sentinel Lymph Node Oriented Treatment Strategy for Gallbladder Cancer" by Yasukawa and colleagues provides a valuable evaluation of the role of sentinel lymph node oriented treatment strategy for gallbladder cancer. As noted in the review, minor corrections are needed prior to consideration of final acceptance of the manuscript in PLOS One.

2. In ethics statement in the manuscript and in the online submission form, please provide additional information about the patient records/samples used in your retrospective study. Specifically, please ensure that you have discussed whether all data were fully anonymized before you accessed them and/or whether the IRB or ethics committee waived the requirement for informed consent. If patients provided informed written consent to have data/samples from their medical records used in research, please include this information.

Reviewers' comments:

Reviewer's Responses to Questions

**Comments to the Author**

1. Is the manuscript technically sound, and do the data support the conclusions?

Reviewer #1: Yes

2. Has the statistical analysis been performed appropriately and rigorously? 

Reviewer #1: Yes

3. Have the authors made all data underlying the findings in their manuscript fully available?

Reviewer #1: Yes

4. Is the manuscript presented in an intelligible fashion and written in standard English?

Reviewer #1: No

5. Review Comments to the Author

Reviewer #1: I commend the authors for address a very important question in GBC surgery. The paper did a good job of putting these findings into clinical context. Please see my attachment for grammatical edits and a question about the definition of DFS.

6. PLOS authors have the option to publish the peer review history of their article (what does this mean?). If published, this will include your full peer review and any attached files.

Reviewer #1: **Yes: **Kristen Spencer, DO, MPH

---

## [Author Response · Author response to Decision Letter 0]

12 Jan 2021

January, 4, 2021

Joerg Heber 

Editor-in-Chief

PLoS ONE

Dear Dr. Heber

We are pleased to submit the first revised version of our manuscript “Applicability of a sentinel lymph node-oriented treatment strategy for gallbladder cancer” to PLoS ONE.

.

We have revised the original manuscript in accordance with the reviewers’ comments. Please find the following files attached:

• Response letter including our point-by-point responses to the reviewers’ comments

• Revised manuscript 1: changes are highlighted in yellow

• Manuscript: before change

This manuscript has not been submitted for publication in any other venue. All coauthors have reviewed and approved the re-submission of this manuscript to the Applicability of a sentinel lymph node-oriented treatment strategy for gallbladder cancer, and all coauthors declare no conflicts of interest.

Please address all correspondence to: Koya Yasukawa, MD

Department of Surgery, Shinshu University School of Medicine

3-1-1 Asahi, Matsumoto, Nagano 390-8621, Japan

Phone: +81 263 37 2654

Fax: +81 263 35 1282

E-mail: kouyayasu@shinshu-u.ac.jp

We hope that our paper will now be considered suitable for publication in the Applicability of a sentinel lymph node-oriented treatment strategy for gallbladder cancer, and we look forward to hearing from you concerning your editorial decision.

Yours sincerely,

Koya Yasukawa, MD 

Associate Editor Comments to Author:

Response: We thank the associate editor for this comment. We changed our manuscript, and please check.

2. In ethics statement in the manuscript and in the online submission form, please provide additional information about the patient records/samples used in your retrospective study. Specifically, please ensure that you have discussed whether all data were fully anonymized before you accessed them and/or whether the IRB or ethics committee waived the requirement for informed consent. If patients provided informed written consent to have data/samples from their medical records used in research, please include this information.

Response: We thank the associate editor for this comment. We added the level 2 heading for sub-sections or major sections, Ethics statements. 

In accordance with the reviewer’s comment, we modified the corresponding text in the revised manuscript, as follows:

Pages 7-8, lines 140-146 (new):

 This study was approved by the ethics committee of Shinshu University School of Medicine (approval no. 2020-4558), and the investigation was conducted according to the principles expressed in the Declaration of Helsinki. All patients were provided with complete information about the study and provided their consent for participation, and written informed consent was obtained from all patients before enrolment. The data were analyzed retrospectively and anonymously on the basis of medical records, and the authors did not have access to identifying patient information or direct access to the study participants.

Response: We are sorry for not describing the Data Availability statement. We added that information.

In accordance with the reviewer’s comment, we modified the corresponding text in the revised manuscript, as follows:

Page 18, lines 312-316 (new):

Data Availability

Data cannot be shared publicly because of the point of view of personal information protection. Data are available from the Shinshu Institutional Data Access / Ethics Committee (contact via shinhp@shinshu-u.ac.jp) for researchers who meet the criteria for access to confidential data. The data underlying the results presented in the study are available from (Shinshu University: shinhp@shinshu-u.ac.jp).

 Reviewer(s)' Comments to Author:

 Reviewer: Kristen Spencer, DO, MPH

I commend the authors for address a very important question in GBC surgery. The paper did a good job of putting these findings into clinical context. Please see my attachment for grammatical edits and a question about the definition of DFS.

Response: Thank you for your comments. We sincerely appreciate these insightful comments, in response to which we have revised and made extensive changes to our manuscript. We believe that our manuscript is greatly improved as a result. Our point-by-point responses to the comments are presented below.

Abstract

Line 39- change malignancy to malignancies

We changed.

Line 52-53 reads as incomplete.

We changed as follows: 

Page 3, lines 53-54 (new):

In T3, patients with skipped LN metastasis were significantly higher in T3h (n=11) than those in T3p (n=2) (P<0.001).

Line 87- would say examination of “the” SLN; and change malignancy to malignancies

We changed.

Lines 90-92- major pathways to what? Assume means lymph drainage but would clarify

We thank your advice. 

We changed as follows: 

Page 4, lines 75-77 (new):

Some researchers[9-11] reported that there were two major pathways for lymph drainage (the left oblique pathway to the celiac nodes and the right descending pathway to the superior retropancreaticoduodenal node) and one minor pathway for lymph drainage (the superior mesenteric nodes).

Line 96- Change according to accordingly

We changed.

Line 104- change to patients who did not have data on location or had missing data on location

We changed.

Line 129- would change to underwent laparotomy

We changed.

Line 131- would add “were done” in T2h

We added.

Line 144- add status (“performance status”)

We added.

Line 153- surgical mortality “was” defined

We added.

Line 164- typically DFS is length of time from randomization (not applicable here) to recurrence of tumor OR death. Does not including patients who recurred OR died impact the study?

We are so sorry for not describing accurately regarding DFS in our previous manuscript. We included patients who recurred OR died in this study. 

We changed as follows: 

Page 8, lines 153-154 (new):

DFS was defined as the time from surgery to recurrence, death from any cause, or the final follow-up.

Line 174- were not significantly different

We changed.

Line 178- would take out “the incidence of LN metastasis” as you already reported this above.

We changed as follows: 

Page 9, lines 166-167 (new):

Furthermore, the incidence of postoperative complications differed significantly between T3p and T3h (0.011).

Line 184- what was the difference for these other post-op complications between T3p & T3h?

We thank the reviewer for this comment. Occurrence of posthepatectomy biliary leakage (T3p: n=2, T3h: n=7) and cholangitis (T3p: n=0, T3h: n=4) was significantly higher in T3h than that in T3p, however, no significant difference was observed in the Clavien–Dindo classification19 grade ≥ III complication between the groups.

In accordance with the reviewer’s comment, we modified the corresponding text in the revised manuscript, as follows:

Page 9, lines 167-170 (new):

Occurrence of posthepatectomy biliary leakage (T3p: n=2, T3h: n=7, P=0.021) and cholangitis (T3p: n=0, T3h: n=4, P=0.038) was significantly higher in T3h than that in T3p, however, no significant difference was observed in the Clavien–Dindo classification19 grade ≥ III complication between the groups.

Line 186- remove each

We removed.

Line 202- change “while” to “as compared to”

We changed.

Line 214- on the peritoneal side

We added.

Lines 215- would change “especially among them” to “specifically” as this sets up for the breakdown of liver recurrence vs nodal

We changed.

Line 216- change to “then that of”

We changed.

Line 219- change to significant

We changed.

Line 220- take out “the”

We removed.

Line 234- change the extent to “extensive LND”

We changed.

Line 238- did not mention

We changed.

Line 240- take out “that”

We removed.

Line 245- change “pf” to “of”

We changed.

Line 254- take out “all”

We removed.

Line 256- change to “few reports of a difference in”

We changed.

Line 261- add “an” before independent poor

We added.

Line 262-263- change to something like “research into how GBC metastases through lymph nodes and the extent of LND are important”

We changed.

Line 265- add conducted in or on; & remove “a” after relatively

We added and removed.

Line 274- change to “how skip LN metastasis occurred”

We changed.

---

## [Editor Report · Decision Letter 1]

1 Feb 2021

Applicability of Sentinel Lymph Node Oriented Treatment Strategy for Gallbladder Cancer

PONE-D-20-32884R1

Dear Dr. Yasukawa,

We’re pleased to inform you that your manuscript has been judged scientifically suitable for publication and will be formally accepted for publication once it meets all outstanding technical requirements.

Kind regards,

Mitesh J Borad

Academic Editor

PLOS ONE

---

## [Editor Report · Acceptance letter]

3 Feb 2021

PONE-D-20-32884R1 

Applicability of Sentinel Lymph Node Oriented Treatment Strategy for Gallbladder Cancer 

Dear Dr. Yasukawa:

I'm pleased to inform you that your manuscript has been deemed suitable for publication in PLOS ONE. Congratulations! Your manuscript is now with our production department. 

Kind regards, 

on behalf of

Dr. Mitesh J Borad 

Academic Editor

PLOS ONE